# Trajectories of COVID-19: A longitudinal analysis of many nations and subnational regions

David Burg[1,2,3,4]*, Jesse H. Ausubel[4]

**1** Tel Hai Academic College, Qiryhat Shemona, Israel, **2** Hemdat Academic College, Netivot, Israel,
**3** Ahskelon Academic College, Ashkelon, Israel, **4** Program for the Human Environment, The Rockefeller University, New York, NY, United States of America

\* david@model-lab.ne

## Abstract

The COVID-19 pandemic is the first to be rapidly and sequentially measured by nation-wide PCR community testing for the presence of the viral RNA at a global scale. We take advantage of the novel "natural experiment" where diverse nations and major subnational regions implemented various policies including social distancing and vaccination at different times with different levels of stringency and adherence. Initially, case numbers expand exponentially with doubling times of ~1–2 weeks. In the nations where interventions were not implemented or perhaps lees effectual, case numbers increased exponentially but then stabilized around $10^2$-to-$10^3$ new infections (per $km^2$ built-up area per day). Dynamics under effective interventions were perturbed and infections decayed to low levels. They rebounded concomitantly with the lifting of social distancing policies or pharmaceutical efficacy decline, converging on a stable equilibrium setpoint. Here we deploy a mathematical model which captures this V-shape behavior, incorporating a direct measure of intervention efficacy. Importantly, it allows the derivation of a maximal estimate for the basic reproductive number $R_o$ (mean 1.6–1.8). We were able to test this approach by comparing the approximated "herd immunity" to the vaccination coverage observed that corresponded to rapid declines in community infections during 2021. The estimates reported here agree with the observed phenomena. Moreover, the decay (0.4–0.5) and rebound rates (0.2–0.3) were similar throughout the pandemic and among all the nations and regions studied. Finally, a longitudinal analysis comparing multiple national and regional results provides insights on the underlying epidemiology of SARS-CoV-2 and intervention efficacy, as well as evidence for the existence of an endemic steady state of COVID-19.

## Introduction

Quantitative studies of viral infection in human severe acute respiratory syndrome coronavirus 2 (SARS-CoV-2) infected subjects have been enabled by the massive global deployment of sensitive and rapid PCR testing for detecting viral RNA in infected persons. Data obtained with these procedures have allowed for extensive mathematical modeling of infection dynamics and viral expansion [1]. Indeed, epidemiological modeling of this pandemic has exploded,

**Data Availability Statement:** Data relevant to this study are available from GitHub at https://github.com/Model-Lab-Net/COVID-19.

**Funding:** The author(s) received no specific funding for this work.

**Competing interests:** The authors have declared that no competing interests exist.

though results have been mixed and show how difficult it can be to provide accurate information and predictions, especially in the early stages of the pandemic [2].

COVID-19 cases initially grew exponentially in every nation. Reduction of community infection was initially achieved by non-pharmaceutical and social distancing interventions [3, 4]. The early and drastic social distancing measures undoubtedly curbed viral expansion [5]. However, the underlying biological, environmental and social dynamics were not fundamentally modified, and viral circulation was only temporarily inhibited. National vaccination programs deployed during 2021 were also aimed to block person-to-person infection. These interventions were enacted at different times, with different levels of enforcement, compliance and extent among nations and in major regions within nations. This global "natural experiment" makes the COVID-19 pandemic a unique opportunity to longitudinally model epidemiological dynamics.

COVID-19 modeling is primarily based on the standard SIR model as the foundational tool of mathematical epidemiology and attempts to capture the main characteristics of the complex interplay among the virus, its host and the environment [6]. The theoretical SIR model's solution converges on a logistic-like S-curve trajectory with rapid expansion reaching a peak and declining in one wave [7]. Many much more elaborate models were deployed to study COVID-19 dynamics [8, 9]; however, complexity invokes problems such as overfitting, global optimization, and interpretability. An important feature not reproduced in these models is the existence of a non-trivial dynamical equilibrium setpoint.

The large amount of publicly available quantitative data amassed allowed a surge of mathematical modeling papers and reports during the COVID-19 pandemic. Researchers have published more than 1,100 peer-reviewed papers in less than two years [10], mostly based on alterations to the SIR model, *e.g.*, the SEIR model and other more complex derivatives [11]. Much work has been performed on modeling the waves of infection which spread across the globe [12]. Saldaña et al. reviewed the main types of epidemiological modeling during COVID-19 [13]. In our extensive reading we found no mention of the V-shaped kinetics observed in the infection data during intervention programs, nor attempts to model the possible endemic steady-state. We aim to show that these are critical characteristics of the virus and the social and pharmaceutical reactions to it, and can be exploited to better understand the observed dynamics of the pandemic. We will show that valuable information about the epidemic can be extracted directly from the kinetics observed in the infection data.

A key criterion of epidemic expansion is the basic reproductive number ($R_o$) which represents a disease's transmissibility. Specifically, it is the average number of productive secondary infections arising from one active infectious individual [14]. It is derived from the ratio between the infection and removal rate constants in the SIR or similar models [15]. A bifurcation threshold condition for the occurrence of a sustained epidemic is $R_o \geq 1$, meaning that as $R_o < 1$ the infection will converge on the disease-free state. This is also an indication for "herd immunity" [16, 17]. In contrast to the outcome of a disease-free state, most models in the context of COVID-19 lack the capacity to depict sustained endemic levels of infection.

Estimation of the value of $R_o$ is commonly based on the initial exponential growth rate [18] and the median infectious period [19, 20]. This is clearly an overestimate as it disregards the removal rate of cases [21]. Another problem is it ignores the distinctive infection peak and inherent inevitable negative second derivative predicted by SIR models. Other approximations treat reproductive rates as a function of time during the epidemic and Wallinga & Lipsitch [22] summarize the main methods to calculate this time-dependent "effective" $R$ ($R_e$). A recent review demonstrated that Cori et al. [23] derived an accurate estimate for this parameter [24]. It has also been suggested that a simple Dirac delta distribution can be used as a proxy for $R_e$ [25]. These are important though $R_e$ will fluctuate as a function of the changes in infection

rates as the epidemic develops [26], but further discussion is beyond the scope of this paper. While these track changes in infection rates change over time (*e.g.*, the first derivative) they do not capture the underlying fundamental biological and social interactions.

This paper highlights applicability of mathematical modeling based on the viral dynamics paradigm [27–29]. A notable characteristic of these methodologies is an endemic-like non-trivial, non-zero, infection dynamical steady or equilibrium state. Further, they can directly model the effects of interventions to block transmission of the pathogen throughout the population. Its major advantage is the ability to derive estimations for the values of model parameters directly from the data [30].

We refrain from exploring the dynamics of the COVID-19 virus itself. SARS-CoV-2, the virus that causes COVID-19, is continuously changing and accumulating mutations in its genetic code. Some variants emerge and disappear, while others emerge, spread, and replace previous variants. For the USA, variant proportions are tracked at https://covid.cdc.gov/covid-data-tracker/#variant-proportions. Obviously, the strategies for suppression can interact with the evolution of the virus. We simply assume a virus which is able to evolve so that it can reinfect previously infected individuals.

Publicly available data for COVID-19 were used to characterize the epidemiological dynamics of community infection. The implementation of efficacious social distancing and lockdown interventions instituted across many nations allows the modeling of the dynamics of infection decay and subsequent rebound as interventions were lifted or lose effectiveness. A longitudinal comparison among nations and major subnational regions provides insights into pathogenesis that would be difficult or impossible to obtain in past pandemics.

## Materials and methods

### Epidemiological data

Data for confirmed active infected cases, COVID-19-associated mortality and PCR tests were retrieved from [31]. For most purposes we stop in September 2021 when the widespread availability of self-testing changed the testing regimes and reduces the reliability of some of the relevant time series. Preliminary review shows that the data exhibit two artifacts. First, a weekly cycle is clearly observed with a tendency for more reporting in the middle of the week and less during weekends, sometimes with orders-of-magnitude differences. Second, large inter-day fluctuations are reported, sometimes with differences spanning multiple orders-of-magnitude. While it is common to smooth the data with a moving average, the resulting estimates are highly sensitive to the fitting window, especially with small numbers and the extremely noisy data (up to an order-of-magnitude between days). Therefore, weekly averages were adopted here and calculated from the geometric mean of the daily measurements to stabilize the variance in the data [32].

There is clearly a delay between time of infection and reporting. Incubation times for COVID-19 are 6.2 days and the mean generation interval is 6.7 days, with a concurrent latent period of 3.3 days [33]. Further, there is a lag between infection and detection by lab test with a skewed distribution [34, 35]. While the exact value is unknown, it will only offset the data in time and does not affect the shape of the infection trajectories. Therefore, a ten-day delay is applied here to all confirmed case numbers, only shifting them left in time and not affecting the shape of the data.

### Inclusion criteria

Analyses were performed for nations and major subnational regions with 10-fold mean difference between PCR tests and positive confirmed cases, high GDP (PPP) per capita [36]

indicating the ability to perform an extensive testing program, and approximately one log decrease in infections from peak to minimum rates during interventions. The 45 qualifying units include 24 European nations, Australia and New Zealand, the UK and the four nations constituting the UK, 10 USA states, and four Asian nations.

## Interventions, mobility and vaccination coverage

Dates for national policy intervention initiation and termination are available and collated from numerous sources and the COVID-19 stringency index was accessed from [37]. Even so, compliance was imperfect, and mobility was used as a minimal estimate for the cumulative efficacies of the intervention polices to block community infection. With data downloaded from [38, 39], the magnitude decrease in mobility was calculated between the average weekly mobility pre-intervention and the minimum mobility observed within six weeks. This difference was used in the model fit to provide an initial estimate for the intervention efficacy parameter during the first V-shape decrease in early 2020. The number of doses of vaccines were retrieved from Mathieu et al. [40] and population data from the World Bank [41]; these enable calculation of the percent of the populace vaccinated. To compare countries and regions, data are commonly normalized to population size, such as "per million". However, COVID-19 is strongly dependent on population density [42].Therefore, to alleviate the population density bias, the data were normalized to the built-up area [43, 44].

## Mathematical modeling of COVID-19

The epidemiology of COVID-19 was analyzed here using a mathematical model of viral dynamics, attempting to capture the mechanism of the virus infecting susceptible individuals. The three model compartments include susceptibles ($S$), COVID-19-confirmed individuals ($I$), and free virus particles ($V$). The model assumes that uninfected people are being made available at a constant rate ($\sigma$) and the virus productively infects them with probability $\beta VS$. Detected infected individuals are removed by quarantine at rate $\delta I$. Viral particles are released from infected individuals at rate $pI$ and are inactivated at rate $cV$. These assumptions lead to the coupled nonlinear ordinary differential eqs:

$$\frac{dS}{dt} = \sigma - (1 - \eta(t))\beta VS$$

$$\frac{dI}{dt} = (1 - \eta(t))\beta VS - \delta I \tag{1}$$

$$\frac{dV}{dt} = pI - cV$$

This is the simplest epidemiological model which affords a non-trivial non-zero infection steady state. A global stability analysis can be found here [45]. Table 1 summarizes the model parameters.

Intervention efficacy to block infection, via social distancing, lockdowns and/or vaccination, is parameterized here by $\eta(t)$. Assuming partial and incomplete effectiveness, e.g., $0<\eta<1$, the system will converge on a new lower steady state. The parameter $\sigma$ is usually interpreted as the repopulation rate of $S$, though here it can also indicate the constant availability of new susceptibles to the virus as it diffuses through the population. While $\sigma$ can be expanded in more elaborate terms, e.g., as a function of time or recovered individuals, we demonstrate that a constant value suffices as a first approximation to provide a dynamical endemic steady state.

**Table 1. Model parameter summary.**

| Parameter | Symbol | Units |
|---|---|---|
| Susceptible influx | $\sigma$ | $S \cdot wk^{-1}$ |
| Infection rate constant | $\beta$ | $V^{-1} \cdot wk^{-1}$ |
| Infected removal rate constant | $\delta$ | $wk^{-1}$ |
| Intervention efficacy | $\eta$ | % |
| Virus production rate constant | $p$ | $wk^{-1}$ |
| Virus decay rate constant | $c$ | $wk^{-1}$ |
| Time of intervention implementation | $t_0$ | time |
| Time of intervention efficacy cessation | $t_1$ | time |
| Infection decay rate | $r$ | $wk^{-1}$ |
| Infection rebound rate | $r_1$ | $wk^{-1}$ |
| Basic reproductive number | $R_o$ | -- |

The mean infectious time is $1/\delta$. The average number of virus particles produced during the infectious interval of a single infected person (the burst size) is given by $p/c$. While asymptomatic carriers are thought to be efficient spreaders, they are not included here as no information is available for this group, and we assume as a first approximation that their dynamics are similar with $I$ and probably change in tandem with the confirmed cases.

COVID-19 associated deaths can be thought of as a subset of infected persons. Indeed, death rates appear to be in a quasi-steady state with the infection rates, being consistently 1-2log lower though lagging by 4–6 weeks throughout the period studied. A Granger causality test provides statistical evidence for this observation ($r = 0.95$, $p<0.01$). For analytical simplicity, they are not modeled here explicitly.

Sustained viral propagation ensues if, and only if, the average number of secondary infections that arise from one productively infected person is larger than one ($R_o >1$). This is the basic reproductive number and for Eq (1) it is defined by $R_o = \beta\sigma p/(\delta c)$. The intrinsic growth rate constant, $r$, is solved for by the dominant root of the eq:

$$r^2 + (\delta + c)r + \delta c(1 - R_o) = 0 \qquad (2)$$

However, if $c>>\delta$ and $r$, then this can be simplified to: $r = \delta(R_o - 1)$. When $R_o>1$, then infection rates will initially experience an exponential increase [46].

The model predicts that as the infection grows it decelerates. The infection will converge in damped oscillations to the non-trivial equilibrium:
$\bar{S} = \delta c/\beta p$, $\bar{I} = (R_o - 1)\delta c/(\beta p)$, $\bar{V} = (R_o - 1)(\beta)$. This dynamical steady state is obtained when the number of new infections equals the number of recovering individuals, where every productive infection generates, on average, only one more new secondary infection.

Assuming a quasi-steady state, $i.e.$, the viral dynamics are much more rapid than the epidemiological phenomenon ($p>>c$), then Eq (1) can be reduced to:

$$\frac{dS}{dt} = \sigma - (1 - \eta)\beta' IS$$

$$\frac{dI}{dt} = (1 - \eta)\beta' IS - \delta I \qquad (3)$$

$$\beta' = \beta p/c$$

with no loss of generality for the major trajectories of infection dynamics [47]. This functional

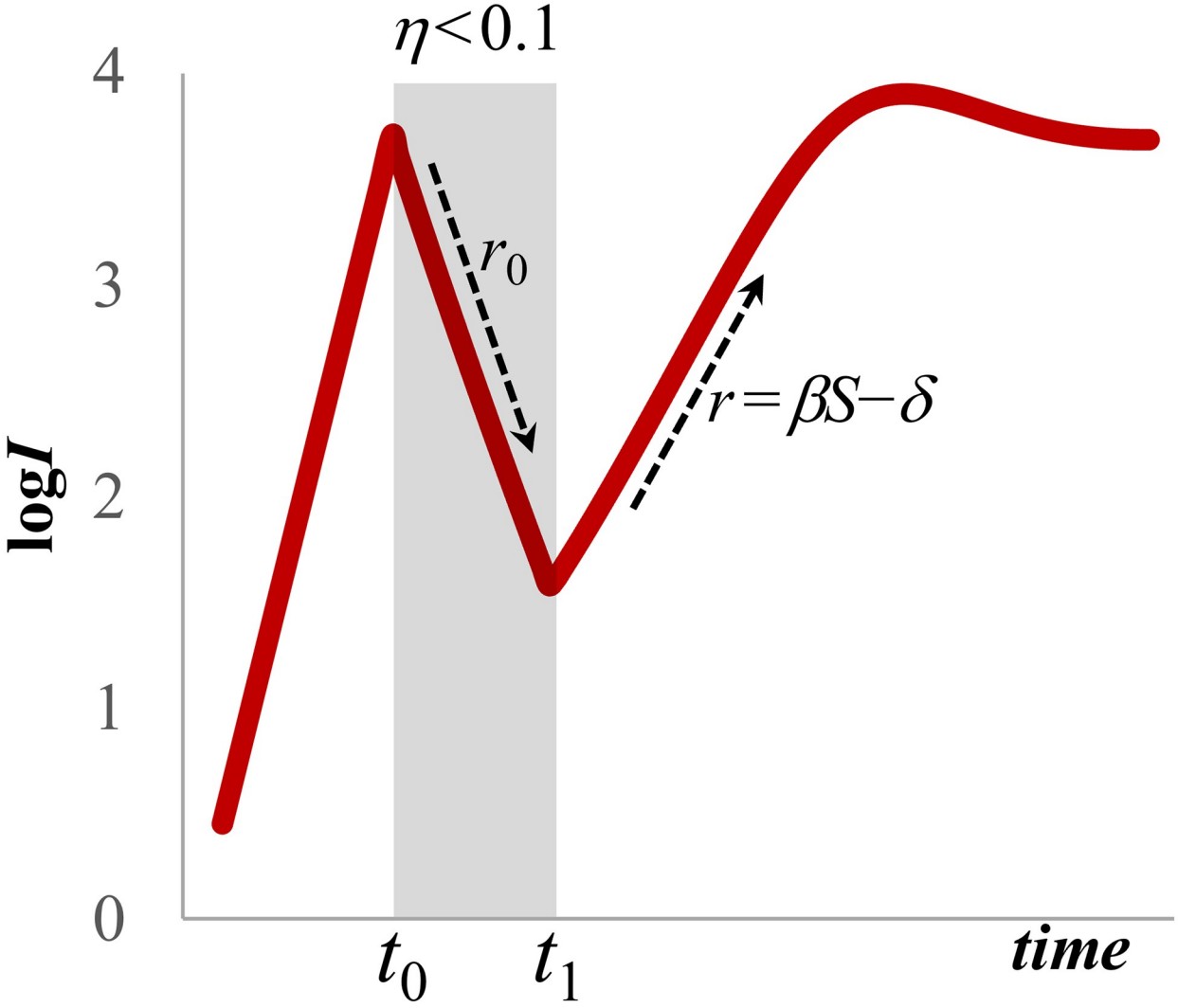

**Fig 1. Epidemiological dynamics under interventions to block infection. Initially, infections rise exponentially (though national COVID-19 testing programs were also ramping up).** During stringent intrvention and effective cessation of viral transmission, between $t_0$ and $t_1$, infection decays exponentially with a half-life of $t_{1/2} = \ln(2)/r_0$, where $r_0$ is derived from the slope of the ln-transformed infection data. This provides a minimal estimate for the value of parameter $\delta$, assuming partial intervention efficacy ($0<\eta<1$). This decay will decelerate reaching a lower steady state. Infections will naturally rebound upon lifting of interventions and/or loss of vaccine efficacy with a doubling time of $t_2 = \ln(2)/r$ and $r$ also calculated from the exponential up-slope. The system will converge with damped oscillations to an elevated infection steady state. This basic pattern will recur as interventions are deployed at different times. Parameter values: $\sigma = 10^4\ S\cdot wk^{-1}$, $\beta = 10^{-5}\ V^{-1}\cdot wk^{-1}$, $\delta = 0.64\cdot wk^{-1}$, $\eta = 70\%$, $t_0 = 16$, $t_1 = 28$.

form has the advantage to decrease model complexity, especially because the viral compartment is less relevant at the community-scale. Exponential decay under interventions to block infection is given by $r_0 = \delta-(1-\eta)\beta'S_0$, where $S_0$ are the number of susceptibles before interventions are implemented ($t_0$). Under highly efficient interventions, *i.e.*, $\eta \rightarrow 1$, then a minimal estimate for $\delta$ can be derived directly from the observed decay half-life of $t_{1/2} = \ln(2)/\delta$ [48, 49]. This model is capable of simulating the dynamics shown in Fig 1.

When interventions are withdrawn, lockdowns are rescinded, other NPI become lax or vaccines become ineffectual, at time $t_1$ then infections will rebound at an exponential rate given

by $r = \beta'S_1 - \delta$, where $S_1$ is the level of available susceptibles at $t_1$. Crucially, $r$ can be obtained directly from the observed slope on the semi-log graph, and its doubling-time is $t_2 = \ln(2)/r$. This expansion in infections will continue in damped oscillations returning to the steady-state.

## Estimation of the basic reproductive ratio

The basic reproductive number is based on a ratio among all five model parameters. However, the paucity of independent knowledge and accurate values for them precludes adequate approximations of $R_o$. To alleviate this, the relationship between the basic reproductive ratio ($R_o$) and the exponential growth rate ($r$) can be recovered such that $R_o = 1 + r(r+\delta+c)/\delta c$. If $r+\delta$ is small compared to $c$, then this approaches:

$$R_o = 1 + r/\delta \tag{4}$$

which can be calculated directly from the exponential slopes, $r_0$ and $r$, as described above.

## Parameter values and statistical analysis

To determine the initial values for model parameters, half-life decay during interventions and rebound doubling-times were calculated from the $\log_n$-transformed data of confirmed cases (weekly geometric means). Optimized values were generated by nonlinear fitting [50], minimizing the objective function $J = \sum_{i=1}^{n} \log(O_i/P_i)$ where $O_i$ and $P_i$ are the observed and expected values, for $n$ datapoints, with the advantage of stabilizing the variance during the fit [32]. Many functional forms for intervention efficacy ($\eta$) can be used but for simplicity, generalizability and as a first approximation:

$$\eta(t) = \begin{cases} \eta, & t_0 < t < t_1 \\ 0, & \text{otherwise} \end{cases}$$

for each intervention wave. The observed decrease in mobility is used here be used as a proxy to estimate its value for each country [51]. Trivially, the proportion of the population needed to be vaccinated in order to block community spread, known as "herd immunity" threshold is [52, 53]:

$$H = 1 - 1/R_o \tag{5}$$

Longitudinal comparisons on the parameter values are performed using the Mann-Whitney u test. 95% confidence intervals, along with their statistical significance, are calculated as appropriate. Model errors (RMS) are reported. Data, simulations and results are available online at: https://github.com/Model-Lab-Net/COVID-19.

## Results

### Dynamics of COVID-19 epidemiology

A preliminary analysis of confirmed COVID-19 cases from 12 nations which did not implement stringent intervention policies, or were unsuccessful at their implementation, indicates widely varying rates and infection levels (Fig 2). By the end of February 2020 these nations had initial infection levels of ~$10^\circ$ cases per km$^2$ with sustained infection doubling times of 1.2–1.7 weeks. Levels increased exponentially for 20±8 weeks and stabilized around a dynamical steady state with fluctuations no larger than 0.5log. Setpoints among these countries fluctuated around 100–400 cases per km$^2$ built-up area per day. Interestingly, South Africa and Armenia

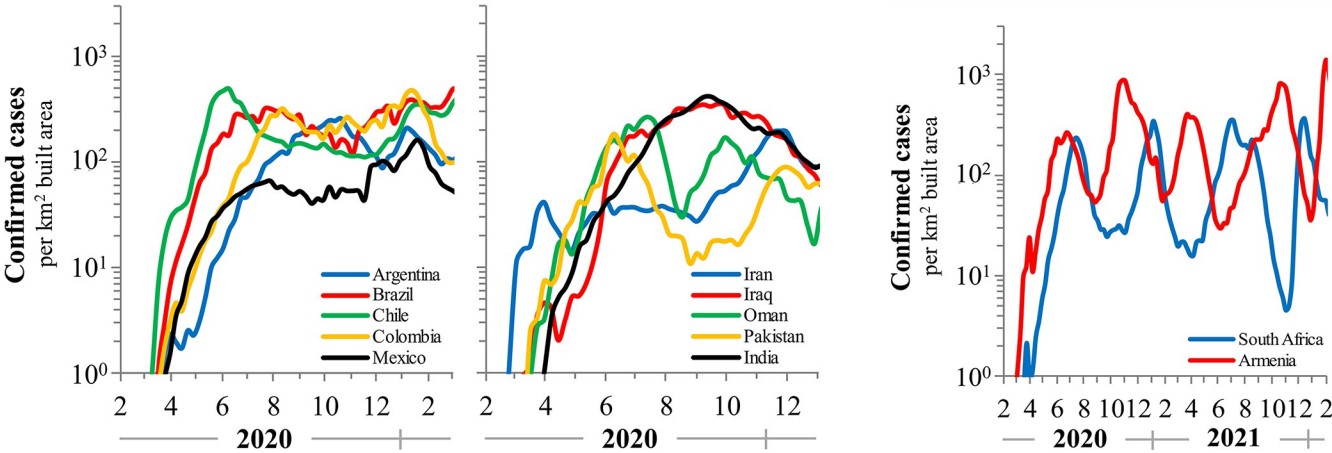

**Fig 2.** A) COVID-19 case levels for 10 nations with no or ineffective interventions increased nearly-exponentially then spontaneously stabilized around 100 cases per km2 built area. B) The Republic of South Africa and Armenia exhibit cycling infection dynamics with spontaneous orbits around a setpoint of approximately 100 cases per km$^2$ built area for 24 months. Data are normalized to built-up area to account for density effects in infection rates.

exhibited spontaneously oscillating kinetics with an amplitude of approximately one order-of-magnitude, perhaps alluding to the existence of a 'limit cycle'. India exhibited one of the largest differences in infection over time, increasing to $10^{2.5}$, declining to $10^{1.5}$ then peaking at $10^3$ before declining spontaneously again to $10^2$ cases per km$^2$ built-up area per day. Because there were no observed effective measures to block COVID-19 spread, the number of confirmed cases attained a dynamical equilibrium around which case numbers fluctuated.

## Dynamics during effective lockdowns

COVID-19 positive case turnover allows analysis of effective social distancing through population-level lockdowns. Non-pharmaceutical means to block new rounds of infections were initially rapid and effectively implemented. Infections begin to decay exponentially 7–10 days after the lockdown policies are implemented, with down slopes of 0.5±0.3 per week and corresponding to half-life values of 2.0±1.1weeks. Infection rates attained nadir within 4–6 weeks with average efficacy of 68% (range: 46–93%), declining 1-2log lower than pre-lockdown case numbers. Confirmed cases rebounded exponentially with doubling times of 2.3-2.6 weeks following the end of severe lockdowns. The trajectory then converged on an empirical equilibrium steady state of approximately $10^2$-$10^3$ cases per km$^2$ built area and with fluctuations less than 0.5log. See Fig 3.

The UK as a whole had, on average, similar dynamical characteristics as its neighbors. However, the observed decay rates during lockdowns were significantly less rapid, leading to differences that will be expanded upon later. While the initial doubling times before lockdowns were similar to other nations and regions, half-lives during lockdowns were nearly twice as rapid, 1.3±0.5 *vs.* 2.0±1.1 weeks. Asian nations, generally, had somewhat different COVID-19 trajectories probably due to the unique measures induced in the included countries here. The Asian rebound rates differed less relative to other countries, though they were more prolonged with some clear oscillatory effects. Additionally, the setpoint infection rates in Japan and South Korea were an order-of-magnitude lower than in Europe. See Fig 3.

The USA is composed of distinct political entities, with large inter-state variation. SARS-CoV-2 surged and waned differently, peaking and ebbing at different times among the various states. Therefore, analyses of COVID-19 for the USA have been done at the state level. Ten states conformed to the inclusion criteria. The US state COVID-19 dynamics were less extreme

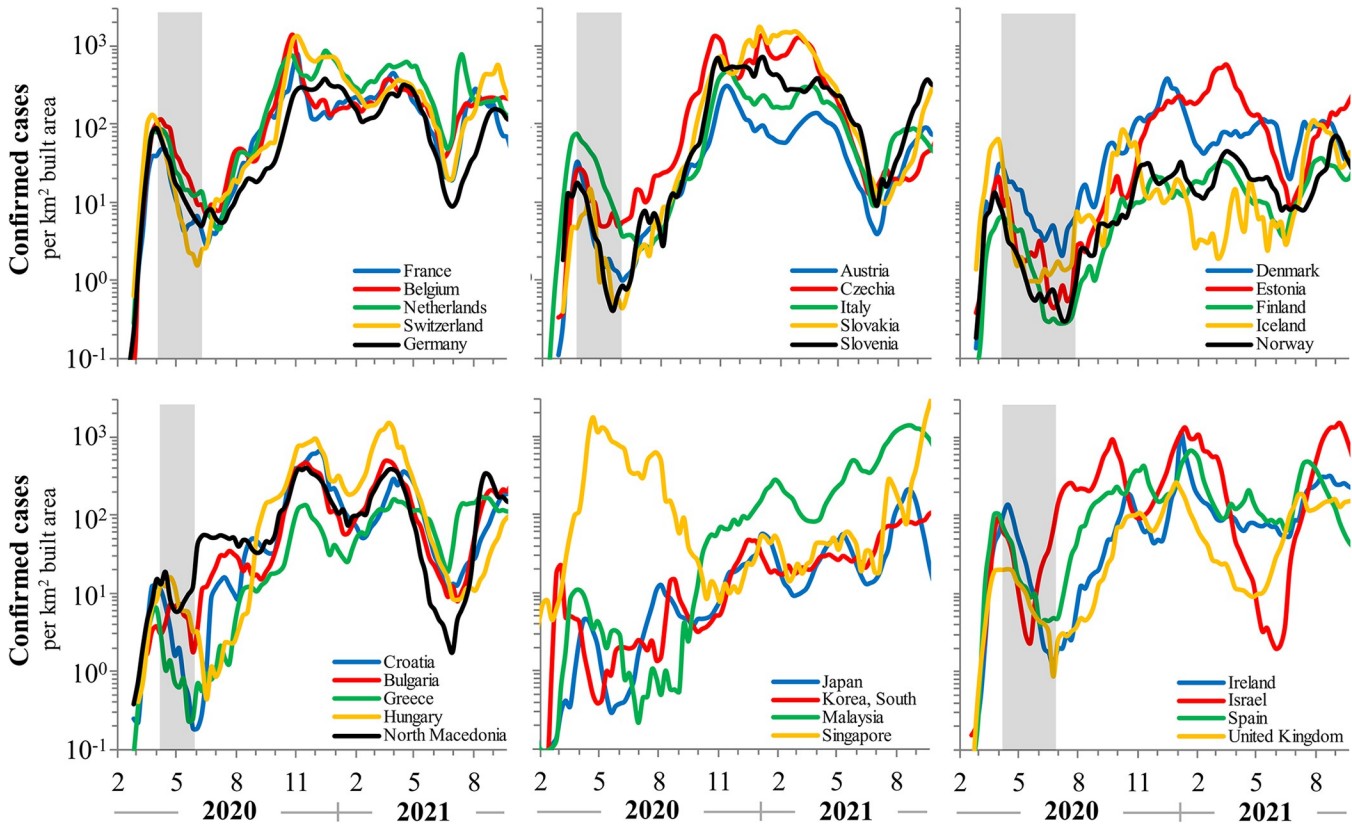

**Fig 3. COVID-19 positive confirmed cases between February 2020 and September 2021.** Data are normalized to built-up area to account for density effects in infection rates. On this scale the recurring patterns become apparent. The exponential decay during lockdowns and following vaccination is clear, as are the geometric rebound trajectories. On this scale the recurring patterns in COVID-19 community diffusion kinetics are undoubtedly evident. Shaded areas indicate the duration of aggressive interventions such as social lockdowns.

with lockdown declines of less than 2log in most states, albeit the up- and down-slopes during lockdowns were comparable with European nations. Four states suffered elevated steady-states approximately one order-of-magnitude higher ($10^{3.2}$–$10^{3.5}$ cases per built-up area per day). See Fig 4.

The earliest, most stringent and prolonged restrictions were implemented in Australia and New Zealand. Confirmed case rates were perturbed to extremely low levels and kept at about 0.5log below the lowest rates achieved in Europe for 35 months, until July 2021. Even so, these strict "Zero COVID" policies were insufficient to completely snuff out community spread. As limits were relaxed, infections surged exponentially with doubling times and equilibrium states comparable to elsewhere, even in the milieu of high vaccination coverage. See Fig 5.

## Modeling of early COVID-19 infection dynamics

The frequent and robust PCR testing for COVID-19 deployed in nations and regions included here allow for the mathematical analyses of infectious persons. Results of the modeling and the parameter values obtained are found in Table 2. The infection dynamics parameter values were obtained from the exponential slopes directly from the data. Initial infection expansion rate constants were 0.5-0.7 per week during February-March 2020, with corresponding doubling times of 1.2-1.6 weeks. Social distancing, lockdowns, and other such interventions resulted in exponential decay of infection rates from the pre-intervention peak values of

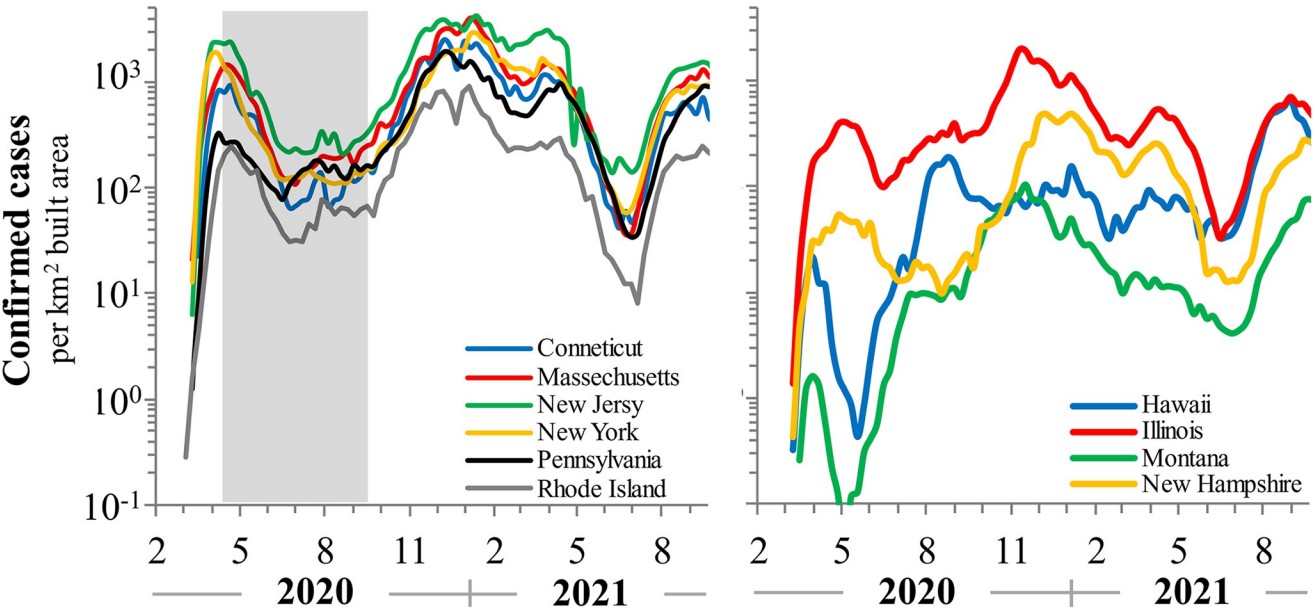

**Fig 4. COVID-19 positive confirmed cases in ten US states conforming to inclusion criteria from February 2020 to September 2021.** More rural and less dense populations have lower COVID-19 infection rates, in general. Data are normalized to built-up area to account for density effects in infection rates.

0.4-0.5 per week with half-life values of 1.7-2.3 weeks. This provides a maximal estimate for the case recovery rate constant parameter ($\delta$).

Infection rates rebounded with doubling times of 2.6–3.7 weeks (range:0.6–4.4 weeks) upon lifting of the extreme social distancing measures. These represent a minimal estimate for $r_0$. This is four-fold less rapid than the initial pre-intervention exponential growth rates. Finally, after 4–12 weeks infections reached a relatively stable setpoint level with values ranging among countries ranging between $10^{1.3}$–$10^{3.4}$ ($CI_{95\%}$: $10^{2.3}$–$10^{2.6}$) cases per $km^2$ built-up area per day. Notably, initial pre-intervention infection rates are significantly correlated with steady state infection levels (PPMCC = 0.41, P = 0.037) alluding to the importance of the intrinsic infection rate and extent of very early viral expansion in the infective dynamical and endemic steady state.

Similar patterns were observed for 10 states in the USA and five nations in Asian regions. Israel implemented a second lockdown intervention during September to November 2020 leading to infections decaying with a half-life of 1.5 weeks and a subsequent rebound with a doubling time of 2.0 weeks; values which are only 15 and 43% more rapid than those during the primary lockdown, respectively. Markedly, not only were decay and rebound slopes among countries of similar magnitude, but they were also similar among infection waves within countries.

## Basic reproductive number ($R_o$)

The analytical approach here contributes insight on the basic reproductive ratio for the community spread of SARS-CoV-2. In the literature reporting on COVID-19, and other epidemics, this is approximated from the initial exponential growth phase [18] and, as noted previously, represents an overestimation because it ignores the $\beta/\delta$ ratio. Here the "natural experiment" of the efficient impedance of viral community spread during the initial phase of the SARS-CoV-2 pandemic allows the use of the empirical rebound up-slope ($r$) and values for

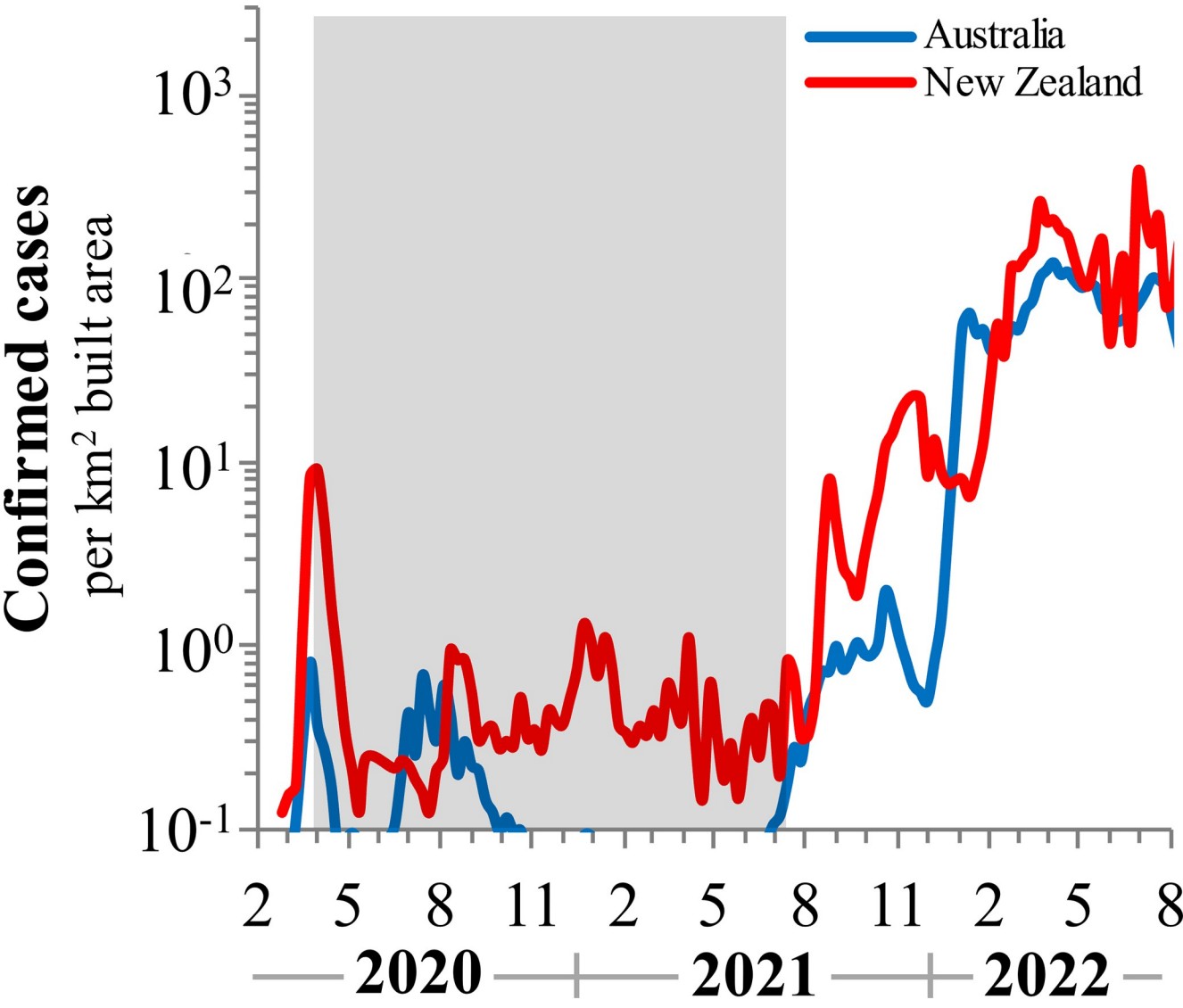

**Fig 5. COVID-19 positive confirmed cases for Australia and New Zealand from February 2020 to September 2021.** The strict "Zero COVID" policies implemented for 35 months kept infection levels at low rates but they rebounded when restrictions were lifted and achieved levels similar to those in Europe. Shaded areas indicate the duration of aggressive interventions. Data are normalized to the built-up area to account for density effects in infection rates.

the recovery/removal rate constant ($\delta$). The estimates for the basic reproductive number are provided in Table 3. Using experimentally established values for $\delta$ (0.4–0.5) from the decay slope during interventions to block viral expansion and ranges for $r$ (0.2–0.3) leads to basic reproductive numbers ranging between 1.4–2.3, narrowing for a $CI_{95\%}$ to 1.6–1.8. From this perspective, active COVID-19 infected individuals would generate approximately 1.7 new secondary infections, on average.

## Herd immunity and inhibition of infection by vaccination

Herd immunity is a threshold value at which new infections cannot perpetuate within the community and is derived from the basic reproductive number. Indeed, nearly all countries which had rapid vaccine rollouts experienced a delayed but rapid exponential decline in case

**Table 2. COVID-19 kinetic characteristics in countries with no effective interventions.**

| Country | Initial growth | Time to steady state | Steady State |
|---|---|---|---|
| | rate $t_2$ | weeks | $\log I \pm SD$ |
| Argentina | 0.2 2.8 | 25 | 2.2 ± 0.3 |
| Armenia* | 0.4 2.0 | 11 | 2.2 ± 0.4 |
| Brazil | 0.6 1.1 | 13 | 2.4 ± 0.2 |
| Chile | 0.4 1.6 | 13 | 2.4 ± 0.3 |
| Colombia | 0.3 2.4 | 21 | 2.4 ± 0.3 |
| Costa Rica | 0.5 1.3 | 26 | 2.4 ± 0.5 |
| Ecuador | 1.3 0.5 | 19 | 1.9 ± 0.1 |
| El Salvador | 0.4 1.7 | 17 | 1.8 ± 0.3 |
| India | 1.1 0.6 | 20 | 3.9 ± 0.5 |
| Iran | 0.2 4.1 | 45 | 2.3 ± 0.4 |
| Iraq | 0.4 1.7 | 22 | 2.5 ± 0.2 |
| Mexico | 0.8 0.8 | 17 | 1.7 ± 0.2 |
| Oman | 0.5 1.5 | 11 | 2.1 ± 0.4 |
| Pakistan | 0.3 2.6 | 25 | 2.3 ± 0.4 |
| Peru | 0.8 0.8 | 16 | 2.5 ± 0.3 |
| S. Africa* | 0.4 1.9 | 16 | 1.8 ± 0.4 |
| **mean** | **0.5 1.7** | **20** | **2.3** |
| **CI$_{95\%}$** | **0.4–0.7 1.2–2.2** | **15–24** | **2.0–2.6** |

*) limit cycle dynamics

numbers with efficacies of 44–99% (CI$_{95\%}$: 64–72). These half-life values following the distribution of SARS-CoV-2 vaccinations (CI$_{95\%}$: 1.3–1.7. Table A1 in S1 Appendix) are similar to those during the early NPI and lockdown interventions. The observed percent of the population vaccinated concomitant with decay in confirmed cases is between 44–55%, based on the nations and regions included here (Table 3). Now it is possible to test the previous calculation of $R_o$, which should be smaller than the observed values. Indeed, the observed "herd immunity" was slightly above the values derived mathematically, as expected from Eq (4), thereby supporting our earlier estimates for the basic reproductive number. Finally, these are clearly lower than reported values for $R_o$ in other studies which seem extremely high.

## Delta variant wave rebound

In June 2021, after the large decrease in COVID-19 following national vaccination programs, COVID-19 cases rebounded spontaneously. The wave was apparently driven by the Delta variant, which became the dominant variant. This rebound was characterized by doubling times of 1.1–1.3 weeks (Table A1 in S1 Appendix). Infections attained average rates similar to those observed prior to vaccination deployment. The decay due to vaccinations and this resurgence both correspond to the trajectories observed in early 2020.

## Discussion

Infection doubling times ($t_2$) and half-life ($t_{½}$) values reveal consistent rates with extremely small variance and narrow range, longitudinally, among all countries analyzed here (Table 2). Mean doubling times for infection levels during the initial exponential phase of the pandemic were 1.0 weeks (CI$_{95\%}$: 0.5–2.0). These were quite robust with a caveat about the rate of deployment of testing regimes.

**Table 3. Optimized COVID-19 model parameter values.**

| | Country | Initial growth | | Decay Slope | | Intervention efficacy | Rebound trajectory | | Steady state infection rate | Reproductive number | Herd immunity | | RMS |
|---|---|---|---|---|---|---|---|---|---|---|---|---|---|
| Europe | | rate | $t_2$ | $\delta$ | $t_{1/2}$ | eta | $r$ | $t_2$ | $\log I \pm SD$ | $R_o$ | obs | exp | |
| | | wk$^{-1}$ | wks | wk$^{-1}$ | wks | % | wk$^{-1}$ | wks | | number | % | % | |
| | Australia | 0.3 | 2.3 | 0.8 | 1.6 | 53 | 0.41 | 1.7 | 1.8 ± 0.1 | 1.2 | 52 | --* | 0.17 |
| | Austria | 0.5 | 1.4 | 0.4 | 1.7 | 71 | 0.3 | 2.6 | 2.0 ± 0.2 | 1.6 | 38 | 25 | 0.19 |
| | Belgium | 0.3 | 1.8 | 0.3 | 2.3 | 66 | 0.3 | 2.3 | 2.3 ± 0.1 | 1.8 | 44 | 35 | 0.13 |
| | Cyprus | 0.2 | 4.1 | 0.4 | 1.9 | 50* | 0.3 | 2.6 | 2.2 ± 0.4 | 1.7 | 41 | 40 | 0.33 |
| | Czechia | 0.6 | 1.1 | 0.5 | 1.5 | 60 | 0.3 | 2.2 | 2.9 ± 0.2 | 1.7 | 41 | 33 | 0.23 |
| | Denmark | 0.3 | 2.1 | 0.2 | 3.9 | 69 | 0.2 | 4.6 | 2.0 ± .02 | 1.9 | 47 | --* | 0.21 |
| | Estonia | 1.3 | 0.5 | 0.4 | 1.8 | 64 | 0.2 | 3.0 | 2.4 ± 0.2 | 1.6 | 38 | 26 | 0.19 |
| | Finland | 1.0 | 0.7 | 0.3 | 2.1 | 51 | 0.3 | 2.7 | 1.2 ± 0.2 | 1.8 | 44 | 22 | 0.15 |
| | France | 0.7 | 2.0 | 0.6 | 1.2 | 79 | 0.3 | 2.5 | 2.3 ± 0.2 | 1.5 | 33 | 33 | 0.18 |
| | Germany | 0.5 | 1.3 | 0.4 | 2.0 | 57 | 0.3 | 2.2 | 2.4 ± 0.2 | 1.9 | 47 | 28 | 0.20 |
| | Greece | 0.4 | 1.6 | 0.2 | 3.0 | 80 | 0.2 | 3.0 | 1.9 ± 0.2 | 2.0 | 50 | 25 | 0.26 |
| | Hungary | 0.9 | 0.8 | 0.5 | 1.5 | 75 | 0.4 | 1.9 | 2.7 ± 0.3 | 1.8 | 44 | 37 | 0.26 |
| | Iceland | 0.7 | 0.9 | 0.9 | 0.7 | 58 | 0.4 | 2.0 | 1.9 ± 0.3 | 1.4 | 29 | --* | 0.33 |
| | Ireland | 0.5 | 1.5 | 0.4 | 1.6 | 76 | 0.3 | 2.7 | 2.0 ± 0.3 | 1.6 | 38 | --* | 0.27 |
| | Israel | 0.5 | 1.3 | 0.7 | 1.0 | 70 | 0.3 | 2.2 | 2.6 ± 0.3 | 1.5 | 33 | 37 | 0.30 |
| | Italy | 0.5 | 1.3 | 0.3 | 2.7 | 93 | 0.3 | 2.2 | 2.4 ± 0.3 | 1.8 | 44 | 29 | 0.17 |
| | Luxembourg | 0.6 | 1.2 | 0.4 | 1.7 | 71 | 0.4 | 1.9 | 2.5 ± 0.1 | 1.9 | 47 | 30 | 0.15 |
| | Netherlands | 0.6 | 1.2 | 0.7 | 1.0 | 67 | 0.3 | 2.0 | 2.7 ± 0.1 | 1.5 | 33 | 27 | 0.20 |
| | New Zealand | 0.6 | 1.1 | 1.5 | 0.5 | 50 | 0.2 | 3.1 | 2.0 ± 0.3 | 1.7 | 40 | --* | 0.33 |
| | Norway | 0.4 | 1.8 | 0.4 | 1.8 | 56 | 0.2 | 3.3 | 1.3 ± 0.2 | 1.5 | 33 | 22 | 0.19 |
| | Slovenia | 0.7 | 0.9 | 0.6 | 1.1 | 73 | 0.3 | 2.6 | 2.6 ± 0.1 | 1.4 | 29 | 36 | 0.19 |
| | Slovakia | 0.8 | 0.9 | 1.1 | 0.6 | 60 | 0.4 | 1.9 | 2.9 ± 0.2 | 1.4 | 29 | 28 | 0.20 |
| | Spain | 0.4 | 1.7 | 0.3 | 2.7 | 93 | 0.3 | 2.0 | 2.3 ± 0.3 | 2.3 | 57 | 63 | 0.29 |
| | Switzerland | 0.4 | 1.7 | 0.5 | 1.4 | 57 | 0.3 | 2.8 | 2.6 ± 0.3 | 1.5 | 33 | 27 | 0.18 |
| | **UK** | 0.5 | 1.5 | 0.2 | 2.9 | 72 | 0.3 | 2.5 | 3.3 ± 0.2 | 2.2 | 55 | 22 | 0.14 |
| | England | 0.6 | 1.1 | 0.3 | 2.4 | 74 | 0.3 | 2.3 | 3.5 ± 0.2 | 2.1 | 52 | 22 | 0.12 |
| | Wales | 0.4 | 1.8 | 0.3 | 2.3 | 75 | 0.4 | 2.0 | 3.2 ± 0.2 | 2.1 | 52 | 38 | 0.14 |
| | Scotland | 0.8 | 0.9 | 0.4 | 1.6 | 72 | 0.2 | 3.7 | 2.7 ± 0.3 | 1.4 | 29 | 31 | 0.17 |
| | N. Ireland | 1.1 | 0.6 | 0.4 | 2.0 | 67 | 0.3 | 2.4 | 2.8 ± 0.2 | 1.8 | 44 | 16 | 0.18 |
| | **mean** | **0.6** | **1.4** | **0.5** | **2.0** | **68** | **0.3** | **2.5** | **2.4** | **1.7** | **41** | **28** | **0.21** |
| | **SD** | **0.3** | **0.7** | **0.3** | **1.1** | **11** | **0.1** | **0.7** | **0.3** | **0.3** | **8** | **12** | **0.06** |
| Asia | Japan | 0.6 | 1.3 | 0.5 | 1.3 | 80 | 0.4 | 1.9 | 1.3 ± 0.4 | 1.8 | 44 | 46 | 0.12 |
| | Malaysia | 0.5 | 1.5 | 0.6 | 1.2 | 72 | 0.1 | 6.5 | 2.1 ± 0.2 | 1.2 | 14 | 51 | 0.27 |
| | Singapore | 0.7 | 1.0 | 0.4 | 1.9 | 73 | 0.3 | 2.2 | 3.8 ± 0.4 | 1.8 | 44 | --* | 0.27 |
| | S. Korea | 2.0 | 0.4 | 1.0 | 0.7 | 46 | 0.2 | 3.5 | 1.4 ± 0.1 | 1.2 | 16 | --* | 0.23 |
| | **mean** | **1.0** | **1.0** | **0.6** | **1.3** | **68** | **0.3** | **3.5** | **2.3** | **1.6** | **39** | **29** | **0.21** |
| | **SD** | **0.7** | **0.5** | **0.3** | **0.5** | **15** | **0.1** | **2.1** | **0.3** | **0.4** | **13** | **11** | **0.07** |
| US | Connecticut | 0.6 | 1.1 | 0.3 | 2.2 | 56 | 0.1 | 6.5 | 2.7 ± 0.2 | 1.6 | 36 | 34 | 0.22 |
| | Hawaii | 0.8 | 0.9 | 0.9 | 0.8 | 85 | 0.3 | 2.5 | 1.9 ± 0.2 | 1.9 | 47 | 32 | 0.32 |
| | Illinois | 0.7 | 1.1 | 0.3 | 2.3 | 63 | 0.1 | 8.8 | 2.7 ± 0.4 | 1.4 | 29 | 26 | 0.19 |
| | Massachusetts | 0.6 | 1.2 | 0.3 | 2.3 | 69 | 0.1 | 6.9 | 2.8 ± 0.2 | 1.5 | 35 | 35 | 0.17 |
| | Montana | 0.7 | 1.0 | 1.0 | 0.7 | 64 | 0.4 | 1.7 | 1.4 ± 0.4 | 1.4 | 31 | 23 | 0.29 |
| | N. Hampshire | 0.7 | 1.0 | 0.4 | 1.7 | 55 | 0.3 | 2.2 | 2.4 ± 0.2 | 1.8 | 50 | 36 | 0.15 |
| | New Jersey | 0.4 | 1.9 | 0.3 | 2.8 | 68 | 0.2 | 4.3 | 3.5 ± 0.1 | 1.7 | 40 | 35 | 0.14 |
| | New York | 0.3 | 2.3 | 0.2 | 2.8 | 70 | 0.2 | 3.3 | 3.4 ± 0.2 | 2.0 | 50 | 34 | 0.11 |
| | Pennsylvania | 0.4 | 1.6 | 0.3 | 4.4 | 60 | 0.1 | 8.0 | 3.2 ± 0.2 | 1.8 | 44 | 32 | 0.11 |
| | Rhode Island | 0.6 | 1.1 | 0.2 | 3.3 | 69 | 0.1 | 6.4 | 3.4 ± 0.3 | 1.8 | 44 | 34 | 0.20 |

*(Continued)*

**Table 3.** (Continued)

| | Country | Initial growth | | Decay Slope | | Intervention efficacy | Rebound trajectory | | Steady state infection rate | Reproductive number | Herd immunity | | RMS |
|---|---|---|---|---|---|---|---|---|---|---|---|---|---|
| | mean | 0.6 | 1.3 | 0.4 | 2.3 | 66 | 0.2 | 5.1 | 2.7 | 1.7 | 41 | 32 | 0.19 |
| | SD | 0.2 | 0.5 | 0.3 | 1.1 | 9 | 0.1 | 2.6 | 0.2 | 0.2 | 8 | 4 | 0.07 |
| | mean | 0.6 | 1.2 | 0.5 | 2.0 | 67 | 0.3 | 3.2 | 2.5 | 1.7 | 40 | | 0.20 |
| | CI$_{95\%}$ | 0.5–0.7 | 1.2–1.6 | 0.4–0.5 | 1.7–2.3 | 64–71 | 0.2–0.3 | 2.6–3.27 | 2.3–2.6 | 1.6–1.8 | 37–43 | | 0.19–0.21 |

*) No rapid decreases in cases observed following vaccination.

**) Data for the fits come from several sources. See Methods section.

SIR-based models assert that infection is acquired by *S* in physical contact with *I*. Here the full model (Eq 1) includes the viral compartment which we interpret as the amount of virions being expelled by the infecteds (*pI*) which infect the susceptibles. However, assuming that *V* turns over more rapidly than *I*, then Eq (1) reduces to Eq (3) (*i.e.*, with $p >> c$ the viral compartment dynamic is limited to the slower dynamics of the infectious compartment). This is the quasi-steady state discussed earlier. Indeed, this is reasonable since COVID-19 generates large amounts of virus and is believed to be short-lived outside the host [54, 55]. Moreover, here we have the advantage to explicitly track the infected compartment vis-à-vis the confirmed cases positive for SARS-CoV-2, which is not usually possible in *in vivo* viral dynamics studies.

Lockdown interventions were extremely effective by inhibiting physical contact and blocking the virus from circulating. Countries with no effective social distancing measures rapidly reached a setpoint equilibrium state. Limiting movement of the population was related to intervention efficacy. Restrictions to travel of 45–93% decreased infection rates by 10-fold or more, leading to an exponential decay of >90% in confirmed cases. Importantly, this was uncorrelated with the minimal infection numbers. More stringent lockdowns do not appear to confer further inhibition to stop viral diffusion and may signify the existence of an optimum in interventions to block COVID-19. The mean associated half-life value during lockdown interventions was 2.0 weeks (CI$_{95\%}$: 1.7–2.4) with no statistically significant difference among the nations and regions studied here. The epidemiological interpretation of this measure is the maximal value for the recovery rate of infected individuals.

As distancing policies were lifted, infections rebounded exponentially as viral diffusion over the social network is no longer perturbed. Intrinsic doubling times can, therefore, be determined empirically by the up-slope on a semi-log graph. The observed doubling time was consistently 2.5±0.7 weeks in European countries. Asian nations included here had values of 3.5 ±2.1 weeks, perhaps owing to their stricter regulations and higher compliance. In the states of the United States the value was even higher at 5.1±2.6 weeks, perhaps alluding to lower compliance.

Taken together, we are able provide a maximal estimate for the basic reproductive number from analysis of the rebound and decay rates in the V-shape dynamics during perturbations on the system to block infections. $R_o$ is overall quite consistent with a mean value of 1.7 (CI$_{95\%}$: 1-6-1.8), due to the invariance of the model parameters. Spain, Greece, and Britain (*i.e.*, England and Wales) were areas of relatively elevated infectivity with values of 2.3, 2.0 and 2.1, respectively. An important outcome of this calculation is the elucidation of the epidemiological "herd immunity" threshold and the novel ability to verify it empirically from the vaccination coverage.

During emergent pandemics, estimates of the basic reproductive number tend to be overestimated. Early COVID-19 studies reported very high values [56, 57]. Our estimates for $R_o$ pertaining to SARS-CoV-2 vary only slightly during waves of COVID-19, which would make sense if the dynamical properties of the infection did not appreciably change. Interestingly, they are comparable to historical influenza pandemics [58] and commensurate with seasonal influenza outbreaks [59]. Although these estimates are substantially lower than those reported elsewhere for COVID-19, they agree with other studies [60].

Vaccination deployment against SARS-CoV-2 had a dramatic effect on infection rates. Confirmed cases decayed exponentially with a mean half-life value with similar rates as during the interventions of social distancing and lockdowns, after achieving the herd immunity threshold. For example, Israel with its early and rapid vaccination program experienced a half-life of 1.0 weeks in confirmed cases once 45% of the population was immunized. This agrees with the prediction given by the approximations for $R_0$ based on Eq (3).

Following the achievement of herd immunity, after approximately 30 weeks, infections spontaneously rebounded again as the delta-variant emerged. The observed escape trajectory was empirically equivalent to the rebound trajectories following the interventions and with doubling times approximately every 1.2±0.3 weeks, similar to the post-intervention rebound doubling times. Interestingly, the Delta variant emerged in every nation included here within 4 weeks, surprising due to the low volume of international travel. Finally, infection rates returned to similar levels as the pre-vaccination setpoint and invariant among the sampled countries.

Although infection rates tended to initially increase exponentially when numbers were low, they quickly saturated to a level of $10^2$–$10^3$ confirmed cases per $km^2$ built-up area per day. This was reached in nearly all nations and regions within 4–6 weeks, even in absence of interventions. Even New Zealand and Australia with strict and highly effective lockdowns rapidly reached this level of infections with the lifting of social distancing measures. Such observations, seen everywhere, suggest a basic, perhaps fundamental, shared epidemiological dynamic and the importance of population density for the spread of SARS-CoV-2 [42, 61, 62].

As we have shown, waves of both infection and suppression can define COVID-19. Our concluding perspective views the infection data decomposed into their wavelet phases and modeled with the generalized multi-logistic model [63]. This approach allows derivation of the saturation level of cases as well as the "characteristic time" (*F044t*) denoting how long the infection takes to increase from 10% to 90% of its extent. While data for many nations and regions resolve neatly into a succession of waves, Israel is unusual in having excellent data for seven waves of infection (so far), as well as companion data about societal responses and suppression for the first five waves. Fig 6 shows the first five infection waves and their durations ranging from 4.4 to 10.6 weeks. The sequence of waves suggests the extremely dynamic interaction of COVID-19, generating new variants, with the social and medical context, including lockdowns, distancing, and vaccines. Predicting new waves remains an unsolved challenge.

To conclude, the dynamical properties of COVID-19 epidemiology are conserved with consistent kinetic patterns with little variation during multiple waves of infection and globally among nations and subnational regions. Nations and regions which implemented interventions sufficient to block community spread effectively experienced a rapid decline in confirmed cases. However, with lifting of interventions, rates rebounded to the previous high infection rates and attained a relatively stable empirical steady state. For COVID-19, societies so far appear to face a choice between relatively high oscillations involving waves of suppression and infection and lesser oscillations around an endemic setpoint. The approach presented here, based on the viral dynamics paradigm, allows derivation of fundamental measures vital to policy such as the basic reproductive number and the magnitude of intervention efficacies.

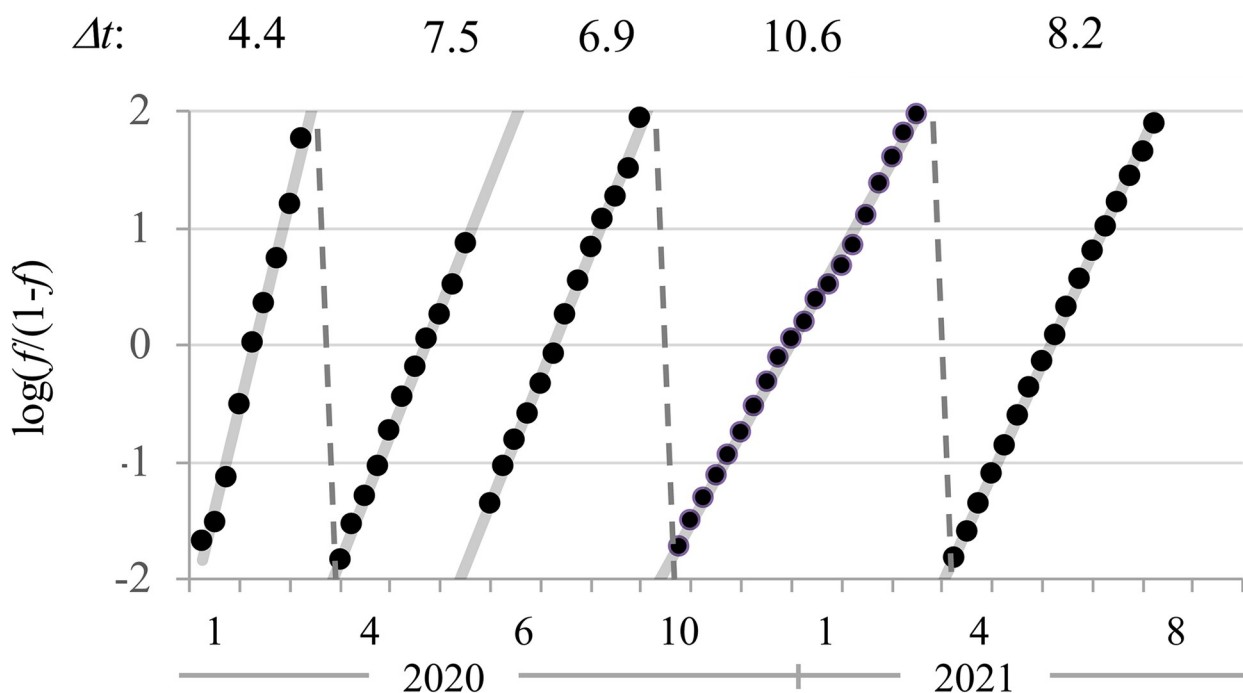

**Fig 6. Logistic curves for first five waves of COVID-19 in Israel and the number of weeks each waves took to run its course.**

Values for $R_o$ derived here of 1.6–1.8 are maximal estimates and lower than other reports. Information on variables of interest for policy normally difficult to obtain is available through this approach and may suggest monitoring strategies efficient for accurate determination of the dynamical properties of future pandemics.

## Supporting information

**S1 Appendix.**
(DOCX)

## Acknowledgments

Yoav Dvir, Mark Y. Stoeckle, and David S. Thaler for important discussions and. Michele Filgatefor editorial assistance.

## Author Contributions

**Conceptualization:** David Burg, Jesse H. Ausubel.

**Data curation:** David Burg.

**Investigation:** David Burg.

**Methodology:** David Burg.

**Writing – original draft:** David Burg, Jesse H. Ausubel.

**Writing – review & editing:** David Burg, Jesse H. Ausubel.

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
