## [Decision Letter · Decision Letter 0]

14 Mar 2023

PONE-D-23-01297Trajectories of COVID-19: a longitudinal analysis of many nations and subnational regionsPLOS ONE

Dear Dr. Burg,

Thank you for submitting your manuscript to PLOS ONE. After careful consideration, we feel that it has merit but does not fully meet PLOS ONE’s publication criteria as it currently stands. Therefore, we invite you to submit a revised version of the manuscript that addresses the points raised during the review process.

Both reviewers agree that the manuscript needs significant revision. Please, when reviewing the manuscript, carefully consider several major issues and several minor issues raised by both reviewers.

We look forward to receiving your revised manuscript.

Kind regards,

Rajnesh Lal, Ph.D

Academic Editor

PLOS ONE

Reviewers' comments:

Reviewer's Responses to Questions

**Comments to the Author**

1. Is the manuscript technically sound, and do the data support the conclusions?

Reviewer #1: Yes

Reviewer #2: Partly

2. Has the statistical analysis been performed appropriately and rigorously? 

Reviewer #1: Yes

Reviewer #2: No

3. Have the authors made all data underlying the findings in their manuscript fully available?

Reviewer #1: No

Reviewer #2: Yes

4. Is the manuscript presented in an intelligible fashion and written in standard English?

Reviewer #1: Yes

Reviewer #2: Yes

5. Review Comments to the Author

Reviewer #1: The authors examined transmission of COVID-19 using data from various countries. They observed the reduction, rebound, and convergence of confirmed cases according to government policies and suggested a mathematical model that could simulate these policies. It has been shown that the estimated values from this model agree with the observed phenomena. In particular, it was a notable result that, contrary to my expectations, the decay rates d and rebound rates r0 were similar in all countries and regions studied during the pandemic. The author has did a good work. hope my comments would help the authors to improve their work.

Major:

1. First, in the introduction section, it would be good to clarify the differences and novelties of this study from related studies. The model used in this study is a basic model used in virus dynamics. This model represents the intracellular spread of the virus in humans. It would be nice to have a detailed reason why this model was applied to the COVID-19 spread simulation. (I recommend introducing related studies, summarizing the results, and writing more prominently the advantages of using this model.)

2. It would be easier to see if there was a table that summarized the description of the parameters, the values, and references used in the model.

3. Regarding comment 2, it would be useful to explain what role a group of free virus particles('V' compartment) play in the COVID-19 spread model in this compartment model. In the general SIR model, it is assumed that an individual belonging to group S is infected by contact with an individual belonging to group I. In this study, however, it is not. The strength of this study is the use of the virus dynamics model for the spread of COVID-19, so it is necessary to explain how the compartment used in the model applies to the spread of COVID-19 in order to emphasize this. (It is a basic model, but it would be nice to describe the dynamics of this model in the Appendix or provide related references.)

4. Line 119 “uninfected people are being made available at a constant rate (\\sigma)”.

Why is the number of uninfected people continuously increasing in the sentence above? (In general, sir-based models assume population growth at birth.)

5. How did you get the lockdown efficacy from Table 2 on page 12? This appears to be related to \\eta in eq(1). However, this \\eta represents not only lockdown, but overall intervention efficacy such as social distancing and vaccination. If the intervention efficacy is interpreted by limiting it to Lockdown, the estimated values in Table 2 are the optimal control solutions for \\eta?

Minor:

1. It would be good to add a diagram representing Eq(1) (lines 124-126).

2. Github address on line 191 is not connected.

3. In Table A1 of Appendix, the post-vax steady state is shown as blank.

4. Since the figures in appendix are in black and white, it is difficult to distinguish between countries.

Positive:

1. I like table2, which summarizes the results for many countries.

2. Figure 6 shows the waves of COVID-19 in Israel well, and at the same time it is good to grasp the "characteristic time" at a glance.

Reviewer #2: The manuscript considers an interesting topic about COVID-19 for several countries in the world. However, this paper is not easy to understand due to errors in writing. The quality of the figures is not high standard and hard to get the information described. The following are some of the errors or unclear descriptions needed to improve.

Remove the dot at the end of title.

Line 120 on page 5: “Deaths can be thought of as a subset …and neglected for the purpose of this study.” If the deaths are significant, I do not think we can neglect. Can you provide evidence from the data you have about this?

Line 136 on page 5: r^2+(δ+c)r+δc(1-R_0 ) should be an equation following the text.

Line 158 on page 6: No Figure 1 is shown here but it is shown at the end. What are the initial values used in this figure?

Line 152: what is t_0? From Figure 1, it is the time exponential decay starting. Please give an explanation.

Line 153 on page 5: insert a space between δ and “can”.

Line 180 on page 7: what is “(Berkeley Madonna v8)”?

Line 196, page 8: the lines used in Fig. 2 for different countries are not readable and time intervals are not the same. The first two are between March 2020 and March 2021 but the last one is between March 2020 and February 2022. When we compare the varying rates, we may use the same time intervals.

6. PLOS authors have the option to publish the peer review history of their article (what does this mean?). If published, this will include your full peer review and any attached files.

Reviewer #1: No

Reviewer #2: No

---

## [Author Response · Author response to Decision Letter 0]

5 May 2023

We thank the reviewers for many helpful comments. Below we respond to each issue.

The file "Response to Reviewers" is formatted for better clarity.

Reviewer #1:

Major:

1. First, in the introduction section, it would be good to clarify the differences and novelties of this study from related studies. The model used in this study is a basic model used in virus dynamics. This model represents the intracellular spread of the virus in humans. It would be nice to have a detailed reason why this model was applied to the COVID-19 spread simulation. (I recommend introducing related studies, summarizing the results, and writing more prominently the advantages of using this model.)

Response: Thank you for this important comment. We added a paragraph to the introduction to distinguish better the contribution of our research to the literature. We added references to 4 papers which deal with the large modeling endeavor during COVID-19. Our aim, however, is not a review paper (and PLOS One is not for review papers), so our summary is brief.

2. It would be easier to see if there was a table that summarized the description of the parameters, the values, and references used in the model.

Response: We added a table summarizing the parameters.

3. Regarding comment 2, it would be useful to explain what role a group of free virus particles ('V' compartment) play in the COVID-19 spread model in this compartment model.

In the general SIR model, it is assumed that an individual belonging to group S is infected by contact with an individua l belonging to group I. In this study, however, it is not. The strength of this study is the use of the virus dynamics model for the spread of COVID-19, so it is necessary to explain how the compartment used in the model applies to the spread of COVID-19 in order to emphasize this.

Response: We agree and added a paragraph to the beginning of the Discussion section to highlight this. To recap, the virus compartment is important for viral diffusion. However, under certain assumptions (e.g., p>>c is large), then the main model dynamics can be reduced to a function of S and I. This is the quasi-steady state.

(It is a basic model, but it would be nice to describe the dynamics of this model in the Appendix or provide related references.)

Response: We agree. We therefore cite in the text: 

De Leenheer P, Smith HL (2003). Virus Dynamics: A Global Analysis. SIAM J Appl Math.;63: 1313–1327, which provides a full analysis of the dynamics of Eq.1, and we moved the reference to a more prominent place in the text (directly after Eq.1).

4. Line 119 “uninfected people are being made available at a constant rate (\\sigma)”.

Why is the number of uninfected people continuously increasing in the sentence above? (In general, sir-based models assume population growth at birth.)

Response: The reviewer is astute to recognize tan issue here. Sigma is indeed a constant rate of influx of susceptible persons. One interpretation is: The virus may be regional and geographic diffusion (not analyzed here) could allow a constant availability of persons susceptible to the virus. Alternately, sigma could be a function. As an initial approach to the problem, we believe a constant parameter which allows a steady state suffices.

5. How did you get the lockdown efficacy from Table 2 on page 12? This appears to be related to \\eta in eq(1). However, this \\eta represents not only lockdown, but overall intervention efficacy such as social distancing and vaccination. If the intervention efficacy is interpreted by limiting it to Lockdown, the estimated values in Table 2 are the optimal control solutions for \\eta?

Response: Another good point. We updated the text to rename “lockdown efficacy” as “intervention efficacy.” The phrase was used as a shorthand for infection suppression during the large decreases of infections concomitant with the lockdowns in early 2020. We have made the needed changes to the manuscript.

Indeed, eta is the efficacy of whatever intervention is implemented. For an initial estimate for the interventions which decreased infections in early 2020, which appear concomitant with the lockdowns, we used public data downloaded from Apple and Google [34,35]. The magnitude decrease in mobility was calculated between the average weekly mobility pre-lockdown and the minimum mobility observed within 6 weeks. This is a minimal estimate for eta in this interval.

In early 2021 infections decreased exponentially in many countries, concomitant with vaccinations, and the value for eta was estimated by the fitting algorithm.

Finally, our interest is in obtaining values of r0 and r1 to capture the main trajectories and calculate R0 rather than in the specific mechanisms and policies of interventions and their relative efficacies. 

In the section on "Methods- Lockdown interventions, mobility and vaccination coverage" we use mobility data as a minimal estimate for the cumulative effects of all interventions to block community infection early in 2020 (line 109-110). However, it seems reasonable that eta is in large part due to severe lockdowns because the exponential decays began concomitantly with those lockdowns in all countries analyzed here, and viral diffusion recommenced following the lifting of the lockdowns. While other interventions (distancing, masking, hygiene, etc.) may impede contraction of the virus, the decay rates suggest that none would have an effect of the same magnitude. When social distancing policies were rescinded in 2022, no exponential increases of >1log were observed. Much research will surely try to dissect the effectiveness of different measures.

Minor:

1. It would be good to add a diagram representing Eq(1) (lines 124-126).

Response: Figure 1 was a qualitative illustration for the dynamics of Eq.1. We replaced it with a numerical simulation and provide the parameter values used to derive it.

2. Github address on line 191 is not connected.

Response: We neglected to make the repo public. It is now available. Apologies.

3. In Table A1 of Appendix, the post-vax steady state is shown as blank.

Response: Fixed

4. Since the figures in appendix are in black and white, it is difficult to distinguish between countries.

Response: Fixed, now in colors.

 

Reviewer #2: 

The manuscript considers an interesting topic about COVID-19 for several countries in the world. However, this paper is not easy to understand due to errors in writing. 

Response: No other readers have commented to us (both native English speakers) about language difficulties. We hope that the 20+ specific improvements made because of the two reviews help readers. It is clear that both reviewers understood the central arguments in the paper.

The quality of the figures is not high standard and hard to get the information described. 

Response: Improved, including addition of colors. Please note that we cannot be responsible for changes performed by PLOS servers. Therefore, we have not removed the figures from the text in the version.

The following are some of the errors or unclear descriptions needed to improve:

Remove the dot at the end of title.

Response: Fixed.

Line 120 on page 5: “Deaths can be thought of as a subset …and neglected for the purpose of this study.” If the deaths are significant, I do not think we can neglect. Can you provide evidence from the data you have about this?

Response: Deaths are observed to be 1-2log lower and lag 4-6 weeks behind infections. They seem to be in a quasi-steady state with infections. A Granger causality analysis provides the statistical evidence for this. We do not mean that deaths are unimportant, only that the focus on cases suffices for our argument. We added a few words to explain better our focus on cases. 

Line 136 on page 5: r^2+(δ+c)r+δc(1-R_0 ) should be an equation following the text.

Response: Fixed.

Line 158 on page 6: No Figure 1 is shown here but it is shown at the end. What are the initial values used in this figure?

Response: We have moved the reference for Figure 1 closer to the Figure itself. Also, we have replaced it with a numerical simulation, and the parameter values are included in the caption.

Line 152: what is t_0? From Figure 1, it is the time exponential decay starting. Please give an explanation.

Response: We added "…before social distancing measures are implemented (t0)."

Line 153 on page 5: insert a space between δ and “can”.

Response: There is a space there but added one more.

Line 180 on page 7: what is “(Berkeley Madonna v8)”?

Response: It is a nonlinear ODE fitting software package. We replaced it with a citation.

Line 196, page 8: the lines used in Fig. 2 for different countries are not readable and time intervals are not the same. The first two are between March 2020 and March 2021 but the last one is between March 2020 and February 2022. When we compare the varying rates, we may use the same time intervals.

Response: We have tried to make the Figure clearer. For clarity we split the first group of countries from South Africa and Armenia (there was also a typing mistake fixed now). The calculated rates are presented in Table 1 for comparison.

---

## [Decision Letter · Decision Letter 1]

31 May 2023

PONE-D-23-01297R1Trajectories of COVID-19: a longitudinal analysis of many nations and subnational regionsPLOS ONE

Dear Dr. Burg,

Thank you for submitting your manuscript to PLOS ONE. After careful consideration, we feel that it has merit but does not fully meet PLOS ONE’s publication criteria as it currently stands. Therefore, we invite you to submit a revised version of the manuscript that addresses the points raised during the review process.

Please address a minor correction as pointed out by Reviewer 2- "Line 158 on page 6: “The intrinsic growth rate constant, r, is solved for by the dominant root of the equation: r^2+(δ+c)r+δc(1-R_0 )” is incorrect.

Since it is referred to as an equation, it should read "r^2+(δ+c)r+δc(1-R_0 )=0".

We look forward to receiving your revised manuscript.

Kind regards,

Rajnesh Lal, Ph.D

Academic Editor

PLOS ONE

Journal Requirements:

Additional Editor Comments:

Please address a minor correction as pointed out by Reviewer 2- "Line 158 on page 6: “The intrinsic growth rate constant, r, is solved for by the dominant root of the equation: r^2+(δ+c)r+δc(1-R_0 )” is incorrect.

Since it is referred to as an equation, it should read "r^2+(δ+c)r+δc(1-R_0 )=0".

Reviewers' comments:

Reviewer's Responses to Questions

**Comments to the Author**

1. If the authors have adequately addressed your comments raised in a previous round of review and you feel that this manuscript is now acceptable for publication, you may indicate that here to bypass the “Comments to the Author” section, enter your conflict of interest statement in the “Confidential to Editor” section, and submit your "Accept" recommendation.

Reviewer #1: All comments have been addressed

Reviewer #2: All comments have been addressed

2. Is the manuscript technically sound, and do the data support the conclusions?

Reviewer #1: Yes

Reviewer #2: Yes

3. Has the statistical analysis been performed appropriately and rigorously? 

Reviewer #1: Yes

Reviewer #2: Yes

4. Have the authors made all data underlying the findings in their manuscript fully available?

Reviewer #1: Yes

Reviewer #2: Yes

5. Is the manuscript presented in an intelligible fashion and written in standard English?

Reviewer #1: Yes

Reviewer #2: Yes

6. Review Comments to the Author

Reviewer #1: The authors wonderfully addressed my concerns and have completed a analysis across many countries. I believe it can be published in PLOS ONE.

Reviewer #2: The manuscript considers an interesting topic about COVID-19 for several countries in the world. The revised version is improved in explanations and graphs. Except for the following issue, I did not find anything else needed to be fixed.

Line 158 on page 6: “The intrinsic growth rate constant, r, is solved for by the dominant root of the equation: r^2+(δ+c)r+δc(1-R_0 )” is incorrect. This was one of the comments for the initial submission, but this problem has not been fixed.

7. PLOS authors have the option to publish the peer review history of their article (what does this mean?). If published, this will include your full peer review and any attached files.

Reviewer #1: **Yes: **Youngho Min

Reviewer #2: No

---

## [Author Response · Author response to Decision Letter 1]

7 Jun 2023

We thank the reviewer for his comment. Below is our response.

Reviewer #2:

Minor:

1. Line 158 on page 6: “The intrinsic growth rate constant, r, is solved for by the dominant root of the equation: r^2+(δ+c)r+δc(1-R_0 )” is incorrect. This was one of the comments for the initial submission, but this problem has not been fixed.

Response: We thank the reviewer and apologies for missing this on the first round. It is now corrected:

r^2+(δ+c)r+δc(1-R_0)=0

---

## [Editor Report · Decision Letter 2]

7 Jun 2023

Trajectories of COVID-19: a longitudinal analysis of many nations and subnational regions

PONE-D-23-01297R2

Dear Dr. Burg,

We’re pleased to inform you that your manuscript has been judged scientifically suitable for publication and will be formally accepted for publication once it meets all outstanding technical requirements.

Kind regards,

Rajnesh Lal, Ph.D

Academic Editor

PLOS ONE

---

## [Editor Report · Acceptance letter]

13 Jun 2023

PONE-D-23-01297R2 

Trajectories of COVID-19: a longitudinal analysis of many nations and subnational regions 

Dear Dr. Burg:

I'm pleased to inform you that your manuscript has been deemed suitable for publication in PLOS ONE. Congratulations! Your manuscript is now with our production department. 

Kind regards, 

on behalf of

Dr. Rajnesh Lal 

Academic Editor

PLOS ONE